

# Highly Scalable Geodynamic Simulations with HYTEG

Ponsuganth Ilangovan[1], Nils Kohl[1], and Marcus Mohr[1]

[1]Geophysics, Dept. of Earth and Environmental Sciences, LMU Munich

**Correspondence:** Ponsuganth Ilangovan (p.ilango@lmu.de)

**Abstract.** High-resolution geodynamic simulations of mantle convection are essential to quantitatively assess the complex physical processes driving the large-scale tectonic phenomena that shape Earth's surface. Accurately capturing small-scale features such as unstable thermal boundary layers requires global resolution on the order of 1 km, which renders traditional sparse matrix methods impractical due to prohibitive memory demands and low arithmetic intensity. Matrix-free methods offer

a scalable alternative, enabling the solution of large-scale linear systems efficiently. In this work, we leverage the matrix-free Finite Element framework HYTEG to conduct large-scale geodynamic simulations that incorporate realistic physical models. We validate the framework through a combination of convergence studies and geophysical benchmarks. These include verifying the convergence rates of Finite Element solutions against analytical solutions and through community benchmarks, including test cases with temperature-dependent and nonlinear rheologies. Our scalability studies demonstrate excellent performance,

scaling up to problems with about $10^{11}$ unknowns in the Stokes system.

## 1 Introduction

Major geological activitiy such as earthquakes and volcanoes can ultimately be attributed to convection processes in the Earth's mantle. The latter gets heated from below by the Earth's core generating density differences that trigger movement of the rocky material in the mantle over geological times scales and in turn drive geologic events (Davies, 1999). In addition, heat production

resulting from the decay of radioactive materials and frictional stresses due to the movement of material also contribute to the thermal state of the mantle.

Creating an accurate model of the Earth's mantle representing this convective process is an essential component for an improved understanding of the geologic processes that shape the surface of our planet, such as the formation of mountain chains and oceanic trenches through plate tectonics, or the release of accumulated stresses during inter-plate earthquakes. Retrodic-

tions of past mantle flow, see e.g. Colli et al. (2018), allow to reconstruct dynamic topography, essential for determining the sedimentation records and to better constrain the mantle's rheology which known is only qualitatively.

On the large timescales involved the mantle can be modelled as a fluid. The high viscosities and small velocities lead to an extremely small Reynolds number, which allows to neglect the inertials terms in the Navier-Stokes equations and assume an instantenous force balance. Hence, mantle convection models are usually based on a generalised form of the Stokes equations.

Historically simulations originally employed the Boussinesq approximation (Oberbeck, 1879; Boussinesq, 1903) to describe the convective process. However, refined models that also take the compressibility of mantle material into account, such as the



(truncated) anelastic liquid apprixomation ((T)ALA), see e.g. Gassmöller et al. (2020) for an overview. A general framework for simulating mantle convection should provide the means to implement various different formulations depending on the predilections of its users.

Since the 1980s various codes for the simulation of mantle convection in 2D and 3D settings have been designed in the community. See Baumgardner (1985); Blankenbach et al. (1989); Moresi and Solomatov (1995); Tan et al. (2006) for examples. Modern variants would be ASPECT (Bangerth et al., 2023), built on top of the FE (Finite Element) framework deal.II and approaches employing Firedrake as underlying FE framework (Davies et al., 2022; Ham et al., 2023).

Simulation of mantle convection is a computationally extremely challenging undertaking. The width of the Earth's mantle is on the order of $3,000\,\text{km}$, while features of interest, such as the thermal boundary layers below the surface and above the Core-Mantle Boundary (CMB), resulting short wavelength asthenosphere dynamics (Brown et al., 2022), or rising plumes and sinking slabs are small scale features of a few km. Resolving them, hence, requires fine resolutions leading to large linear systems of equations. These often need to be solved multiple times per time-step, of which one often has to perform on the order of thousands as one wants to simulate millions of years of Earth's history. Mantle convection codes, thus, have been candidates to be run on supercomputers early on, see e.g. (Bunge and Baumgardner, 1995).

However, not only its sheer size is problematic. Also the strong local viscosity variations that develop in mantle convection models pose a challenge for the solution process and finding optimal iterative solvers is an ongoing research topic, see e.g. Clevenger and Heister (2021).

Classical FE analysis relies on assembling the system matrix of the linear system of equations and then working in terms of numerical linear algebra. However, such traditional matrix-based methods are not well-suited for truly large-scale high-performance computing, due to their immense memory requirements and performance penalizing low arithmetic intensity.

In this work, we present studies performed with the mantle convection code TerraNeo, see Bauer et al. (2020); Ilangovan et al. (2024), designed to run models at extremely fine resolutions. TerraNeo is based on the matrix-free FE framework HYTEG (Kohl et al., 2019, 2024) and the associated automatic code-generator HOG (HyTeG Operator Generator) (Böhm et al., 2025). Our aim here is threefold. We are going to present results for various benchmark settings that are commonly employed in the community to assess the usability and correctness of mantle convection codes (Zhong et al., 2008; Davies et al., 2013, 2022; Euen et al., 2023), to show these problems can successfully be solved with TerraNeo. As part of this we will also evaluate certain model simplifications and adaptions that can potentially improve performance in an HPC (High Performance Computing) context. The third aspect is to demonstrate the scalability of TerraNeo when simulating at a peak global resolution of 3 km resulting in $\simeq 10^{11}$ degrees of freedom (DoFs).

The remainder of this paper starts with a description of the model used to represent convection in the Earth's mantle, its spatial and temporal discretisation as well as some simplifications for a HPC setting. This is followed by a brief overview on HYTEG and HOG and the numerical solution process we employed, before we come to the central part of the paper that consists of a section on benchmarking and validation.





## 2  Model

On geological timescales the mantle behaves like a highly viscous fluid. Its movement can, thus, be described by the Navier-Stokes equations. Due to the large values of viscosity ($> 10^{19}\,\mathrm{Pa\,s}$), the Ekman number (ratio of viscous to Coriolis forces) becomes very large, while the Reynolds number (ratio of inertial to viscous forces) becomes very small (Schubert et al., 2001). This allows to neglect Coriolis and inertial forces, and conservation of mass and balance of forces can be modelled by the Stokes equations alone. The major driving force of the system is buoyancy, which results from density differences in the mantle. These are mostly due to temperature gradients. Our model can, thus, be written as

$$-\nabla \cdot \boldsymbol{\tau} + \nabla p = \rho \mathbf{g} \tag{1}$$

$$\frac{\partial \rho}{\partial t} + \nabla \cdot (\rho \mathbf{u}) = 0 \tag{2}$$

where $\mathbf{u}$ represents velocity, $\rho$ density, $\eta$ viscosity, $p$ pressure, $\mathbf{g}$ the gravity vector and $\boldsymbol{\tau} = \eta \left( \nabla \mathbf{u} + \nabla \mathbf{u}^{\top} \right) - \frac{2}{3}\eta(\nabla \cdot \mathbf{u})\mathbf{I}$ the deviatoric stress tensor. This problem is posed on a thick spherical shell

$$\Omega = \Omega_{r_{\min}, r_{\max}} := \left\{ \boldsymbol{x} \in \mathbb{R}^3 \ \middle| \ r_{\min} < \|\boldsymbol{x}\|_2 < r_{\max} \right\} \ ,$$

used to represent the geometry of Earth's mantle.

It is common practice in the community to work with a reformulation of the problem which involves a hydrostatic reference state. One then considers deviations from this reference state instead of the total physical quantities. In the following we will denote by $\bar{\square}$ a radial reference profile for quantity $\square$ and by $\square'$ deviations from the latter. We can then e.g. express pressure as $p = \bar{p} + p'$, where the hydrostatic pressure $\bar{p}$ satisfies $\nabla \bar{p} = \bar{\rho} \mathbf{g}$. As a next step we approximate total density by a first order Taylor expansion

$$\rho(p, T) \simeq \bar{\rho}(r) \left[ 1 + \bar{\kappa}_T(r) p' - \bar{\alpha}(r) T' \right] \ . \tag{3}$$

Here $r$ denotes a point's distance from the center of the problem domain $\Omega$, while $\bar{\kappa}_T$ and $\bar{\alpha}$ denote isothermal compressibility and thermal expansivity. Equation (3) constitutes one part of the so called anelastic liquid approximation (ALA) (Gough, 1969). The other is to neglect the density variations $\rho'$ in the continuity equation eq. (2), as these can be assumed to be small compared to $\bar{\rho}$. As the latter is constant in time, this also removes the time derivative from the equation, and, thus, also the potential for sound waves in the model. A fact to which the approximation owes its name. By further dropping the pressure dependence in eq. (3), one arrives at the so called truncated anaelastic liquid approximation (TALA)

$$\rho(p, T) \simeq \bar{\rho}(r) \left[ 1 - \bar{\alpha}(r) T' \right] \ . \tag{4}$$

With this the Stokes part of our model obtains its final form

$$-\nabla \cdot \boldsymbol{\tau} + \nabla p' = \bar{\rho} \bar{\alpha} T' \hat{\boldsymbol{r}} \tag{5}$$

$$\nabla \cdot (\bar{\rho} \mathbf{u}) = 0 \ , \tag{6}$$



where $\hat{\boldsymbol{r}}$ is a radial unit vector.

In addition to this, conservation of energy in the mantle needs to be considered (McKenzie and Jarvis, 1980). This can be written in terms of temperature $T$ as primary quantity and couples to the Stokes system. Under the assumptions of the TALA, deviations of density from the reference profile $\bar{\rho}$ can be neglected and one obtains

$$\bar{\rho}\bar{c}_p \left( \frac{\partial T}{\partial t} - \nabla\left(k\nabla T\right)\right) = \boldsymbol{\tau}:\dot{\boldsymbol{\epsilon}} + H + \bar{\alpha}T\mathbf{u}\cdot\nabla p \ . \tag{7}$$

Here $\bar{c}_p$ denotes the specific heat capacity, $H$ the internal heat generation and $k$ the thermal conductivity. The equation is of advection-diffusion type, and the terms on the right-hand side describe, in this order, shear-heating, internal heating (through radioactive decay) and adiabatic heating.

Under the assumption that the pressure deviations from the hydrostatic reference are very small ($\bar{p} \gg p'$) and using again $\nabla\bar{p} = \bar{\rho}\mathbf{g}$, we end up with the energy equation in its final form

$$\frac{\partial T}{\partial t} + \mathbf{u}\cdot\nabla T - \frac{1}{\bar{\rho}\bar{c}_p}\nabla\cdot\left(k\nabla T\right) = \frac{1}{\bar{\rho}\bar{c}_p}\boldsymbol{\tau}:\dot{\boldsymbol{\epsilon}} + \frac{\bar{\alpha}}{\bar{c}_p}\left(\mathbf{u}\cdot\mathbf{g}\right)T + H \tag{8}$$

In most parts of the mantle advective transport is the dominating process, exceeding the effectivness of diffusive transport by several orders of magnitude (Peclet number Pe $\simeq 10^3$).

The model is complemented by choosing initial conditions for temperature, from which one can derive an initial velocity field and and initial pressure by solving the Stokes equations for the resulting buoyancy term. Additionally one needs to impose boundary conditions for $\boldsymbol{u}$ and $T$ on the surface ($r = r_{\text{max}}$) and the CMB ($r = r_{\text{min}}$). Temperature boundary conditions are usually of Dirichlet type, while for velocity different choices are possible. Typically in a mantle convection model one requires

$$
\begin{aligned}
\boldsymbol{u}\cdot\hat{\boldsymbol{r}} &= 0 && \text{on } \delta\Omega \\
\boldsymbol{u}\cdot\boldsymbol{t} &= \boldsymbol{u}_{\text{tangential}} && \text{for } r = r_{\text{max}} \\
\boldsymbol{t}\cdot\boldsymbol{\tau}\cdot\hat{\boldsymbol{r}} &= 0 && \text{for } r = r_{\text{min}}
\end{aligned}
\tag{9}
$$

The first of these constitues a no-outflow condition. The tangential velocity component is taken from plate-tectonic reconstructions. In the final condition $\boldsymbol{t}$ represents any vector in the tangential plane of the boundary point. Thus, it requires that at the CMB the tangential shear-stress vanishes. Combined with no-outflow this constitutes a free-slip condition. Physically it is motivated by the fact that below the CMB lies the outer core, composed of molten iron on which the mantle rocks can freely move.

## 2.1 Space Discretization Method

We employ the Finite Element Method (FEM) for determining an approximate solution of the Stokes problem as well as for discretising the spatial derivatives in the energy equation, apart from the advection term, see section 2.2. Let $\mathcal{T}$ be a triangulation of the problem domain $\Omega$, i.e. a splitting into non-overlapping triangles in 2D or tetrahedrons in 3D. Then the discrete version





of the weak form of the Stokes problem becomes

$$\int_\Omega \boldsymbol{\tau}(\mathbf{u}_h) : \nabla \mathbf{v}_h d\Omega - \int_\Omega p_h \nabla \cdot \mathbf{v}_h d\Omega = \int_\Omega \mathbf{f} \cdot \mathbf{v}_h d\Omega \tag{10}$$

$$-\int_\Omega \nabla \cdot (\bar{\rho} \mathbf{u}_h) q_h d\Omega = 0, \tag{11}$$

Here $\mathbf{f}$ is the right-hand side (RHS) from eq. (5) and $(\boldsymbol{u}_h, p_h)$ are the sought for approximate velocity and pressure solutions, while $(\boldsymbol{v}_h, q_h)$ are test functions. These are chosen from finite-dimensional spaces, commonly e.g. $\boldsymbol{u}_h, \boldsymbol{v}_h \in V_h \subset (H_1(\Omega))^3$ and $p_h, q_h \in Q_h \subset L_2^0(\Omega)$, where the latter denotes the space of all square-integrable functions with zero mean. Note that eq. (10) does not contain any surface integrals for the considered boundary conditions eq. (9). More details on the derivation of the variational formulation including boundary conditions can e.g. be found in Burstedde et al. (2013).

In the case of a free-slip boundary condition, cf. eq. (9), we treat it as a natural boundary condition and only remove the radial velocity component from the test and trial functions to satisfy the no-outflow component. In the context of a curved domain this requires special consideration and we follow the approach by Engelman et al. (1982).

A large variety of different mixed FE function pairs have been suggested in the literature for the Stokes problem, see e.g. Terrel et al. (2012). For our experiments in this study, we have opted for the standard $P_2 - P_1$ Taylor-Hood pair. This is not only
inf-sup stable, but also one of the two choices advocated for in the comparison done by Thieulot and Bangerth (2022).

## 2.2   Time Discretization Method

Finding a stable and accurate time discretization of the energy equation is tricky in our setting as it is strongly advection dominated. A common technique that avoids unphysical oscillations in the numerical solution is the Streamline Upwind Petrov Galerkin (SUPG) approach, which adds a consistent artificial diffusion in the direction of the velocity streamline. This resolves
the problem, but requires heuristic tuning of additional parameters for weighting the stabilization term. Another alternative would be entropy-viscosity stabilisation, see Euen et al. (2023) for details and a comparison to SUPG.

Here, we instead use a Eulerian-Lagrangian approach, in the sense that we treat the advective part of the equation with the Modified Method of Characteristics (MMOC). The latter is based on the fact that in the case of pure advection temperature remains constant along the characteristic curves of the PDE. For full details and benchmarks, see Kohl et al. (2022). Here we
only give a brief overview.

In the MMOC we initialise virtual particles at the locations of the DoFs of the FE temperature field at time $t + \delta t$. We then use an explicit Runge-Kutta scheme of $4^{\text{th}}$ order to trace each particle back in time to its departure point $x_{\text{dept}}(t)$ at the previous time-step $t$. At these positions we evaluate the old temperature field to obtain an advected intermediate temperature $\hat{T}_h(x_{\text{dept}}(t), t + \delta t)$. The intermediate advected temperature is used in further time discretization of the FE weak form of the energy equation.
We discretize the temperature $T_h$ in space with a quadratic $P_2$ FE function, hence by choosing an appropriate test function $s_h$, we can write the weak form with implicit Euler time discretization as,

$$\int_\Omega \frac{T_h(t + \delta t) - \hat{T}_h(t)}{\delta t} s_h d\Omega + \int_\Omega k \nabla T_h(t + \delta t) \cdot \nabla s_h d\Omega - \int_\Omega \alpha (\mathbf{u}_h \cdot \mathbf{g}) T_h(t + \delta t) s_h d\Omega = \int_\Omega f_T(t + \delta t) s_h d\Omega. \tag{12}$$



Here the term $f_T$ conflates the internal heating and shear-heating parts of eq. (8). Note that the latter depends on velocity, and in case of temperature-dependent viscosity models also on $T$. The linear system resulting from eq. (12) can easily be solved

with the help of a Krylov subspace solver. In our experiments we use either Conjugate Gradient or GMRES for this.

Once the temperature at time $(t + \delta t)$ is obtained, we use the same to setup the buoyancy term and solve the Stokes system, thereby obtaining the corresponding velocity $\mathbf{u}(t + \delta t)$.

An aspect that we have not mentioned so far is that solving the individual ODEs for the trajectories of the virtual particles requires to provide the velocity field $\mathbf{u}_h$ at the micro time-steps of the Runge-Kutta method in $[t, t + \delta t]$. In practice this is not

known. We use a predictor-corrector approach in our implementation. In a first step, we simply use the temperature from the previous time-step (constant extrapolation). Once we have solved for the new temperature and velocity at $(t + \delta t)$, we redo the step by linearly interpolating between $\mathbf{u}_h(t)$ and $\mathbf{u}_h(t + \delta t)$ and computing new values for $T_h$ and $\mathbf{u}_h$ at $(t + \delta t)$. In the case of a more pronounced coupling, e.g. in the case of a velocity dependent viscosity, we can repeat this process multiple times.

While experiments in Kohl et al. (2022) indicate that with the MMOC one is not necessarily restricted by the Courant-

Friedrichs-Lewy (CFL) condition, we have taken a conservative approach here and still use it to guide our choice of time-step length. Given a threshold $C_{\mathrm{CFL}} > 0$, the maximal magnitude of velocity field in the domain $u_{\max}$, and maximal FE element length $h$, we get for the CFL time-step length,

$$dt_{\mathrm{CFL}} = C_{\mathrm{CFL}} \frac{h}{u_{\max}} \; . \tag{13}$$

### 2.3 Simplifying Formulations for HPC

The most computationally intensive part of the simulation process is solving the Stokes system. Hence any simplifying approximation or modification is desirable given that it is consistent with the physics of the simulations. In this section, we give a brief of some methods that are tested in section 4.2.1.

#### 2.3.1 Frozen velocity

Using a compressible flow formulation, the density field in the continuity equation (2) becomes spatially varying. This then

spoils the symmetry of the resulting saddle point linear system arising from the FE discretization, as the bilinear forms for the gradient and divergence part are no longer identical. Applying the product rule of the divergence operator one can, in a first step, re-write eq. (6) as

$$\nabla \cdot \mathbf{u} = -\frac{\nabla \rho}{\rho} \cdot \mathbf{u}, \tag{14}$$

and modify the weak form of the TALA formulation, specifically eq. (11) to become

$$\int_{\Omega} (\nabla \cdot \mathbf{u}_h) q_h d\Omega = -\int_{\Omega} \frac{\nabla \bar{\rho}}{\bar{\rho}} \cdot \mathbf{u}_h q_h d\Omega. \tag{15}$$

When solving the Stokes system at time $t_{n+1} = t + \delta t$, one can, as a second step, "freeze" the velocity on the right-hand side to be $u_h(t_n)$, which retains symmetry between the two bilinear forms.





### 2.3.2 Frozen divergence

The full stress tensor $\boldsymbol{\tau}$ contains a divergence part $\frac{2}{3}\eta\left(\nabla\cdot\mathbf{u}\right)\boldsymbol{I}$ which is non-zero in the compressible cases and, when the divergence of $\boldsymbol{\tau}$ is taken, results in a grad-div term. Additionally this also increases the complexity for the core kernel functions which perform matrix-vector operations. We treat this term similar to the frozen velocity case, i.e. by moving it to the RHS of momentum equation eq. (1) and evaluating it with the velocity field from the previous time-step:

$$-\nabla\cdot\left[\eta\left(\nabla\mathbf{u}(t_{n+1})+\nabla\mathbf{u}^{\top}(t_{n+1})\right)\right]+\nabla p(t_{n+1})=\mathbf{f}+\frac{2}{3}\nabla\left[\eta\nabla\cdot\mathbf{u}(t_n)\right] \tag{16}$$

Hence, we obtain an extra forcing term on the right-hand side, but retain the stress tensor in the form of the incompressible setting. In this work we only use this approach for a benchmark with compressible a Stokes system on an unit square in section 4.2.1. More experiments and mathematical analysis would be required to fully validate the effectiveness of the approach and its influence on accuracy.

## 3 Finite Element Framework

Given the desired global resolution of $\simeq 1$ km, holding the data required to solve the resulting linear systems in memory becomes challenging. If we consider a FE coefficient vector with about a trillion ($10^{12}$) entries (which approximately corresponds to the number of elements in such a mesh), then the size of the FE coefficient vector is about 8 TB when working in double precision. Comparing this to the total amount of about 1 PB of main memory of the Hawk supercomputer (66th in Top500 as of Nov '24) that was used for experiments in this paper, we can see that we can fit only 125 such vectors in memory. Since we typically cannot access the entire machine at once, and since this does not include the memory required for the system matrices yet (which will require storing 10-100 or more non-zeros per row) such simulations can only be performed with significantly more main memory or with numerical methods with smaller memory demands.

Since memory access also is relatively slow and, thus, often a performance-limiting factor the better solution is to not form the matrices explicitly and employ matrix-free methods (Brown, 2010; Kronbichler and Kormann, 2012; May et al., 2015; Kohl and Rüde, 2022). This of course restricts the set of solvers that can be implemented to those that work with matrix-free compute kernels, but enables the solution of linear systems at the extreme scale. In this work, we build on the framework HYTEG for parallel, matrix-free FEM simulations and the associated code generation framework HOG for architecture optimized matrix-free compute kernels. Both together have been shown to allow efficient solution of systems at tera scale ($\simeq 10^{12}$ DoFs) (Kohl and Rüde, 2022; Böhm et al., 2025).

### 3.1 HYTEG and HOG

HYBRID TETRAHEDRAL GRIDS (HYTEG) builds upon an unstructured coarse mesh composed of triangles in 2D and tetrahedrons in 3D that captures the basic problem geometry. The elements of the coarse mesh are split into their geometric primitives (vertices, edges, faces and cells), which we denote as macro-primitives. On these we perform structured refinement to reach a



desired mesh resolutions, thereby generating micro-elements. Data associated with micro-entities is stored in the corresponding macro-entities, which act as containers and also form the smallest units for parallelisation in HYTEG, which is based on
the distributed memory paradigm and implemented via the Message Passing Interface (MPI). The refinement process results in a hierarchy of meshes and is, thus, particularly well-suited for the development of efficient geometric multigrid methods. The block-structuredness of the resulting mesh allows using data-structures without indirections, which is beneficial in multiple aspects, e.g. memory access, and avoids the need to store coordinate and topology information on micro-entities, as these can be computed on-the-fly. Additionally it supports implementation of various optimisations, such as e.g. cache-friendly loop
strategies within the matrix-free compute kernels, which are not possible on fully unstructured FE meshes. In order to ease performance portability and testing of new optimisations the core compute kernels such as operator-application (i.e. the analogue of a matrix-vector multiplication in a matrix-free setting) or computation of inverse-diagonals are not implemented manually, but auto-generated via the HYTEG OPERATOR GENERATOR (HOG). This also significantly simplifies addition of new PDE problems and their bilinear forms. For more details on HYTEG and HOG we refer the reader to Kohl et al. (2019, 2024); Böhm
et al. (2025).

## 3.2    Mesh refinement

HYTEG can import arbitrary tetrahedral meshes in GMSH format and also supports inline mesh generators for standard domains. Among these also one for our primary target domain, the thick spherical shell. Construction of a coarse mesh for the latter is based on the icosahedral meshing approach, see Baumgardner and Frederickson (1985); Davies et al. (2013). Here
an icosahedron is mapped onto the unit sphere, resulting in 20 spherical triangles. To these one applies midpoint refinement with geodetic arcs. Once this tangential surface mesh is fine enough it is radially extended to cover $\Omega_{r_{\min}, r_{\max}}$. This results in spherical prismatoids, each of which can be split into three tetrahadrons. To this tetrahedral mesh the standard refinement process of HYTEG is applied, which recursively splits each triangle into four sub-triangles and each tetrahedron into eight sub-tetrahedra.
However, this standard refinement will not improve the domain approximation, due to the curvature of the latter. To resolve this issue, HYTEG employs a blending map, which maps the micro-elements of the refined triangulation onto the actual problem domain. We denote the former as computational domain $\Omega_{\text{comp}}$ and the latter as physical domain $\Omega_{\text{phys}}$, see Bauer et al. (2017) for further details. From an FE perspective HYTEG in such a case employs two mappings. One from the reference element to the micro-element on the computational domain, which is then mapped onto the computational domain. This has to be
accounted for in the element integrals and the gradients of the shape functions. Denoting by $J_A$ the Jacobian of the affine mapping onto the computational domain and by $J_B$ that of the blending map we obtain e.g.

$$\nabla \psi_i = \mathcal{J}_B^{-\top} \mathcal{J}_A^{-\top} \nabla \phi_i \tag{17}$$

for an ansatz function $\psi_i$ and its associated shape function $\phi_i$ on the reference element. The blending map needs to be at least a homomorphism globally and a diffeomorphism locally on each macro-primitive. In the case of the icosahedral mesh for





the thick spherical shell such a blending map can be constructed and additionally its inverse is explicitly known. This is of technical importance for the MMOC approach and an advantage over a quadratic isoparametric mapping.

### 3.3 Linear System Solvers

The linear system of equations that arises from the weak form of the Stokes system is a saddle point problem, which is block-structured as

$$
\begin{bmatrix} A & B^\top \\ B & 0 \end{bmatrix} \begin{pmatrix} U \\ P \end{pmatrix} = \begin{pmatrix} f_U \\ f_P \end{pmatrix}.
\tag{18}
$$

In a matrix-free setting, iterative solvers form the only possible choice for computing solutions. HYTEG offers different approaches for solvers and preconditioners, and due to the hierarchy of levels generated through the structured refinement it naturally supports geometric multigrid approaches. Multigrid methods, either algebraic or geometric, feature in most approaches to solve the Stokes system. This is related to the $A$-block, which results from the discretisation of an elliptic operator. Gmeiner et al. (2016) e.g. compared three different iterative solvers, all involving multigrid in some form, for the isoviscous case and found a monolithic multigrid method with inexact Uzawa smoothing to offer the best performance. The smoother can be written as consecutive velocity and pressure system iterations, see e.g. Gaspar et al. (2014) for details. In Kohl and Rüde (2022) it was shown that using this combination in HYTEG makes it possible to achieve extreme scalability and solve for a trillion DoFs in less than a minute.

While the monolithic multigrid approach proves to be extremely scalable, we found it to have robustness issues in cases where viscosity is sharply varying. As this is often the case in geodynamic scenarios we opted for this paper to instead employ a Krylov subspace solver with multigrid preconditioning. To be more precise, we use a Flexible GMRES (FGMRES) method in combination with the following preconditioner $\mathcal{P}$

$$
\mathcal{P} = \begin{bmatrix} \hat{A} & B^\top \\ 0 & -\hat{S} \end{bmatrix}.
\tag{19}
$$

Here $\hat{S}$ approximates the system's Schur complement. A common choice for $\hat{S}$ is to use the pressure mass matrix scaled by the inverse of viscosity, as this is spectrally equivalent to the actual Schur complement, as long as viscosity is not too heterogeneous, see Rudi et al. (2017) and the references therein. The approximation $\hat{S}$ can be constructed as

$$
\hat{S}_{ij} = \int_\Omega \mu^{-1} \phi_i \phi_j d\Omega \ .
\tag{20}
$$

Application of the preconditioner requires multiplication with $\mathcal{P}^{-1}$, and we use multigrid for solving a linear system with $A$ as part of this. Strongly variable coefficients, as in our case viscosity, pose a challenge for geometric multigrid, as without proper handling the convergence may degrade significantly. In a matrix-free setting, the construction of suitable coarse grid operators, e.g. by using a Galerkin Coarse Grid Approximation (GCA), see Trottenberg et al. (2001), is generally difficult or at least expensive (Knapek, 1998).



For this study we have opted for the following approach. We employ a Direct Coarse Grid Approximation (DCA), sometimes
also refered to a rediscretisation, in combination with a homogenization approach for the viscosity. For the latter we first
convert viscosity into an element-wise constant, i.e. $P_0$, function on the finest level by computing an arithmetic average.
This $P_0$ viscosity is then restricted to the coarser levels of the mesh hierarchy by recursively computing the an average of the
viscosity over the child elements on the finer level of a coarse level element (Clevenger and Heister, 2021). We use an arithmetic
average here in most models. For the case of nonlinear rheology we found a harmonic average to work better. Besides its use in
the multigrid part, note that this averaging also has another positive effect. If viscosity exhibits sharp variations in the domain,
pressure becomes inherently discontinuous. Using a Taylor-Hood approach the approximate pressure solution is forced to
be continuous, though. This results in spikes in the solution, which get damped when one employs this averaging approach
(Heister et al., 2017).

### 3.4   Rigid Body Modes

In some benchmarks and models free-slip boundary conditions are imposed on both the inner and outer boundary of either
the thick spherical shell or the annulus. In both cases this results in a non-trivial kernel for the $A$ block of linear system
eq. (18) composed of rigid body modes. This can lead to convergence problems with the iterative solver or the quality of the
approximate solution. To avoid such issues one can employ a nullspace removal algorithm that filters out the rotational mode
components from the residuals and solution vectors during every step of the iterative solver. This, however, will increase the
number of dot products and require additional MPI communication, which in turn can affect the run-time performance of the
solver.

Thus, we follow a different approach and penalize rigid body modes in the velocity field by extending the weak form of the
momentum equation eq. (10) by the term

$$\sum_{d=\{x,y,z\}} \int_\Omega c_{\text{rot}} \left(\mathbf{e}_{\text{rigid}}^d \cdot \mathbf{u}_h\right) \left(\mathbf{e}_{\text{rigid}}^d \cdot \mathbf{v}_h\right) d\Omega. \tag{21}$$

Here, $\mathbf{e}_{\text{rigid}}^d(\mathbf{x})$ denotes the unit vector describing an elementary rotation around the $d$-axis, e.g. in the case of the $z$-axis we
have $\mathbf{e}_{\text{rigid}}^z = [-y/r, x/r, 0]^\top$. The penalty value $c_{\text{rot}}$ is chosen as a fixed global value and tuned by hand. The extra term can
easily be incorporated into the code and optimized by using the HOG code generator. The benefit here is that no extra MPI
communication is needed, since the handling of the rigid body modes happens as part of the compute kernel for operator
application.

## 4   Benchmarking and Validation


In this section, we benchmark HYTEG against various geophysical problems. Initially, we consider mathematical conver-
gence studies, with analytical solutions of the Stokes problem on the thick spherical shell. Then we move to time-dependent
community benchmarks including setups with compressible flow, temperature-dependent and non-linear rheologies. This is a





continuation of the work done in Ilangovan et al. (2024) where several simple benchmark experiments have been conducted
with HYTEG.

## 4.1 Stationary Benchmarks

Although the order of convergence of the FE approximation can be confirmed with the method of manufactured solutions
(Roache, 2001), application scenarios that are relevant to geodynamics often cannot be verified analytically, e.g., convergence
behaviour when a Dirac-delta type forcing is present in the system. Discontinuities (or sharp variations) of material parameters
are common in geodynamic applications. For incompressible Stokes flow on the annulus and spherical shell (semi-)analytical
solutions were derived in e.g. Blinova et al. (2016); Horbach et al. (2019); Kramer et al. (2021). In section 4.1.1 we first
study HYTEG subject to the setup from Kramer et al. (2021), where the authors derive solutions for cases where the forcing
function is in the form of a Dirac delta function. To keep this setup more closer to our geodynamic application (where Dirichlet
boundary conditions are considered on the outer surface), the boundary conditions chosen here is the mixed case (noslip on the
outer boundary, freeslip on the inner boundary). Next in section 4.1.2, we consider the freeslip-freeslip case to test the order
of convergence using the rigid body mode penalty method proposed in section 3.4, for which we consider the smooth forcing
function. As a baseline comparison for both of the cases, we consider the mixed case boundary condition setup with smooth
forcing and present the same together with the $\delta$ function forcing in section 4.1.1.

### 4.1.1 $\delta$-function forcing

Consider the incompressible, isoviscous Stokes system

$$-\nabla \cdot \left(\nabla \mathbf{u} + (\nabla \mathbf{u})^{\top}\right) + \nabla p = f\mathbf{g} \ , \tag{22}$$

$$\nabla \cdot \mathbf{u} = 0 \ . \tag{23}$$

The domain considered is a thick spherical shell with $r_{\min} = 1.22$ and $r_{\max} = 2.22$ so as to keep the aspect ratio of the domain
close to that of Earth's mantle $\frac{r_{\min}}{r_{\max}} \simeq 0.55$ and to keep mantle thickness as 1. Next, we impose the mixed type boundary
condition with no-slip on the outer and free-slip on the inner boundary. The major focus here is to test with the $\delta$ forcing case,
but for baseline comparison, we also consider the smooth forcing. The corresponding forcing functions $f_\delta$ and $f_{\mathrm{smooth}}$ are,

$$f_{\mathrm{smooth}} = \frac{r^k}{R_{\max}^k} \mathcal{Y}_{\ell m}(\theta, \phi) \ , \quad f_\delta = \delta\left(r - r_d\right) \mathcal{Y}_{\ell m}(\theta, \phi) \ , \tag{24}$$

where $\mathcal{Y}_{\ell m}$ is the spherical harmonic function of order $\ell$ degree $m$, and $\delta\left(\cdot\right)$ denotes the Dirac delta function. The analytical
solution for the $\delta$-function case will be a radial function piecewise defined on the two branches $r < r_d$ and $r > r_d$, whilst
satisfying an appropriate interface condition. Previous works have computed solutions for the mixed boundary condition case
and have shown the convergence of the FE solution in HYTEG to the analytical one, with convergence orders expected for
a $P_2 - P_1$ pairing (Ilangovan et al., 2024). Hence, here the main focus is to test convergence of the same with the developed
multigrid solver. The coarse mesh contains 3 layers with nodes present on the outer, inner and the radial layer at $r_d = 1.72$, so




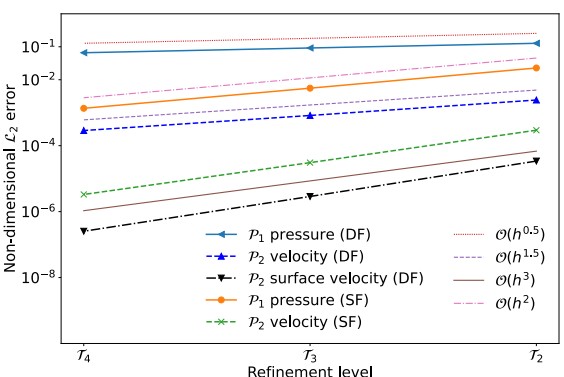
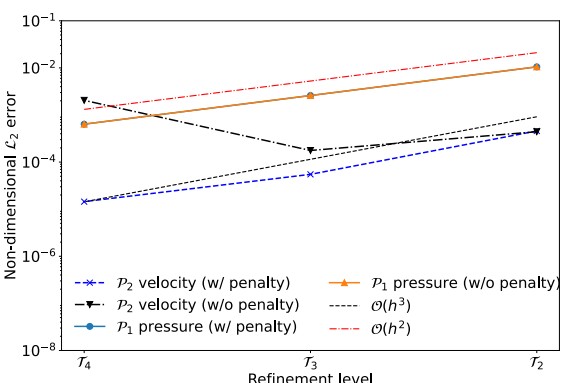

**Figure 1.** (Left) Order of convergence of the $P_2 - P_1$ FE solution to analytical solution on the Spherical Shell with noslip condition on outer and freeslip on the inner surface (mixed case) under the smooth forcing (denoted by SF) and $\delta$-forcing (denoted by DF) function case; (Right) Order of convergence of the $P_2 - P_1$ FE solution to analytical solution on the Spherical Shell with freeslip imposed on both surfaces under the smooth forcing case and tuned rotational mode penalty

as to mark the domain in HYTEG where the $\delta$-function needs to be applied. To apply the delta function, the associated volume

integral in the weak form of the RHS of FE discretized Stokes system, must be transformed into a surface integral. Consider the RHS in the weak form of Stokes system, where the $\delta$-function is then prescribed as,

$$\int_{\Omega} \delta(r - r_d) \mathcal{Y}_{\ell m} \phi_i \mathbf{n} d\Omega = \int_{\Gamma_{r_d}} \mathcal{Y}_{\ell m} \phi_i \mathbf{n} d\Gamma_{r_d}. \tag{25}$$

The convergence study is started at $\mathcal{T}_2$ and considered up to $\mathcal{T}_4$. The FGMRES solver preconditioned with geometric multigrid is used as the linear system solver. The coarsest level for the multigrid solver is set to $\mathcal{T}_2$. From the computed solutions fig. 1

(left), we see that the FE solution converges to the actual analytical solution for the smooth forcing (denoted by SF) with the theoretical order of convergence for a $P_2 - P_1$ Taylor Hood element, i.e. $\mathcal{O}(h^3)$ for velocity and $\mathcal{O}(h^2)$ for pressure. Whereas for the $\delta$-function forcing case, we see from fig. 1 (left, denoted by DF), a deterioration in convergence of both the velocity with $\mathcal{O}(h^{1.5})$ and pressure with $\mathcal{O}(h^{0.5})$. This is supposedly expected as the analytical pressure solution is discontinuous, which is then unable for the $P_1$ FE element to represent accurately (Kramer et al., 2021). Another aspect to note is that, as the delta

function is applied on the center of the domain, the accuracy of velocities on the surface with free-slip condition is achieved with the theoretical convergence order of $\mathcal{O}(h^3)$.

### 4.1.2  Rotational Mode Penalty

Here we focus on the convergence for the case with free-slip boundary condition imposed on both the inner and outer surfaces while prescribing smooth forcing to the Stokes system. The dimensions of the domain considered is same as in the previous

section except, here, the coarse mesh of the spherical shell consists of 2 radial layers which are then structurally refined to





reach the required refinement level. The FGMRES solver preconditioned with geometric multigrid is used as the linear system solver. The convergence study is considered from $\mathcal{T}_2$ up to $\mathcal{T}_4$ of which the coarsest level of the multigrid solver is always $\mathcal{T}_2$. For the comparison with/without the rigid body mode penalty values, the number of FGMRES iterations is kept the same at every refinement level $\mathcal{T}_l$.

First, we note from the previous section that, when a mixed type boundary condition is considered, then the expected order of convergence of $O(h^3)$ for velocity and $O(h^2)$ for pressure is achieved. Whereas, when freeslip is imposed on both surfaces , then the convergence is severely deteriorated when $c_{\text{rot}}$ from section 3.4 is set to 0 (see fig. 1 (right) w/o penalty). An interesting aspect (although expected) to note is that, as seen from fig. 1, the rigid body mode only spoils the velocity solution, whereas the pressure solution achieves the expected convergence irrespectively. The order of convergence is gained back when an

appropriate $c_{\text{rot}}$ value is set to the system (see fig. 1 (right) w/ penalty). There is no theoretical work done on the penalty value under the scope of this study, and this value was handtuned in this work, although roughly behaves as $O(h)$.

## 4.2   Geophysical Benchmarks

In this section, we will consider numerical convection experiments that are commonly used in the geophysics community to benchmark mantle convection frameworks. First in section 4.2.1, we consider a compressible convection simulation under the

TALA approximation on an unit square and test the approximation techniques proposed in section 2.3.1 and section 2.3.2 and verify the resulting Nusselt number against other codes. Then in section 4.2.2, we consider the benchmark from Tosi et al. (2015) with a viscoplastic rheology. Finally, in section 4.2.3, we consider the benchmark from Ratcliff et al. (1996); Zhong et al. (2008) on the spherical shell and compare the radially averaged temperatures and Rayleigh number vs Nusselt number trends reported in other works.

### 4.2.1   Compressible case

We consider the compressible Stokes equations eq. (5) and eq. (6) coupled with the energy equation eq. (8) on the unit square. The goal is to perform an experiment using our methodologies with the setup identical to King et al. (2010) and compare the Nusselt numbers that we achieve on the top boundary. Free-slip boundary condition are imposed on all sides. In addition to diffusion and adiabatic heating/cooling in the energy equation eq. (8), we also consider the shear heating term. The experiment

is started with an initial sinusoidal perturbation,

$$T(\mathbf{x}, t=0) = (1-y) + A_0 \cos \pi x \sin \pi y, \tag{26}$$

with $A_0 = 0.1$, which then induces a single convection cell in the square. An isoviscous, compressible case with $\bar{\rho} = \exp \frac{1-y}{\text{Di}}$ is considered, where Di is the dissipation number taken as $\text{Di} = 0.5$. The unit square mesh contains 12 subdivisions in the $x$ and $y$ direction generating a total of 288 triangles at the coarsest level $\mathcal{T}_0$. The triangles are refined to reach the operating mesh

at level $n$ denoted as $\mathcal{T}_n$. To solve the linear system of equations the FGMRES solver with multigrid as preconditioner is used. The finest level of the multigrid is $\mathcal{T}_n$ and the coarsest level is set at $\mathcal{T}_0$. The time-steps are calculated with the CFL condition. At each time-step, 10 iterations are performed with the FGMRES solver, with two coupling Picard iteration between the Stokes





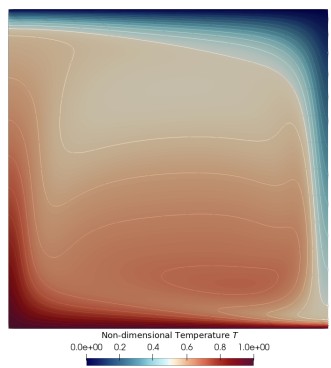
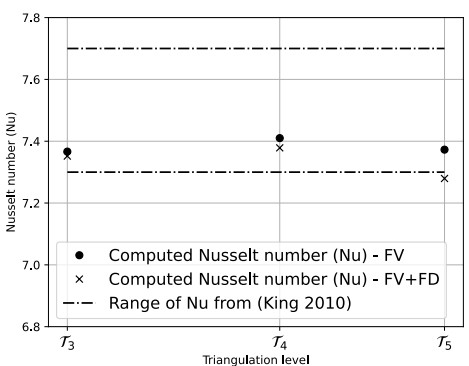

**Figure 2.** (Left) Final Temperature contours of the compressible benchmark on an unit square of the TALA case from King et al. (2010) presented in section 4.2.1 with Ra $= 10^5$ and Di $= 0.5$; (Right) Comparison of Nusselt number values between the Frozen velocity approach (FV, see section 2.3.1) and including Frozen divergence approach (FD, see section 2.3.2) under the setup from section 4.2.1 identical to the Truncated Anelastic Liquid Approximation (TALA) case from King et al. (2010) at Ra $= 10^5$ and Di $= 0.5$.

and energy equations. The system eq. (5), eq. (6) and eq. (8) is non-dimensionalised as done in King et al. (2010), which then introduces the Rayleigh and Dissipation number. The simulation is performed at Rayleigh number Ra $= 10^5$ and run until the temperature reaches a steady state as seen in fig. 2. After this, the Nusselt numbers are computed on the top boundary and checked for convergence. The Nusselt number is computed at steady state as the ratio between heat transported with convection to the heat transported with pure diffusion. By taking a pure diffusion solution as the reference, $T_{\text{ref}}$, the Nusselt number can be computed for the final temperature solution $T$ as,

$$\text{Nu} = \frac{\int_\Gamma \nabla T \cdot \hat{\mathbf{n}} d\Gamma}{\int_\Gamma \nabla T_{\text{ref}} \cdot \hat{\mathbf{n}} d\Gamma}. \tag{27}$$

This simulation is tested with the frozen velocity approach from section 2.3.1 and also with the frozen divergence from section 2.3.2 and the corresponding Nusselt numbers are reported, see fig. 2 (right). In both cases, they reach the expected values as seen from fig. 2 which shows the comparison with the values from other codes.

### 4.2.2 Nonlinear Rheology

Although the rheology of the mantle is not completely known, it is clear that it exhibits a nonlinearity (Boioli et al., 2018). A simple example would be the plate boundaries, where the plates behave plastic, i.e., large stresses (which depend on the velocity $\mathbf{u}$) make the plates exhibit plastic behaviour which decreases viscosities giving rise to weak zones on the plate boundaries. Here, we show a plume rising experiment with a sinusoidal temperature perturbation (eq. (26)) on the unit square with a nonlinear rheology. The setup is identical to case 4 from Tosi et al. (2015), where the viscosity is defined as,

$$\mu = \left( \frac{1}{\mu_{\text{linear}}} + \frac{1}{\mu_{\text{nonlinear}}} \right)^{-1}, \tag{28}$$





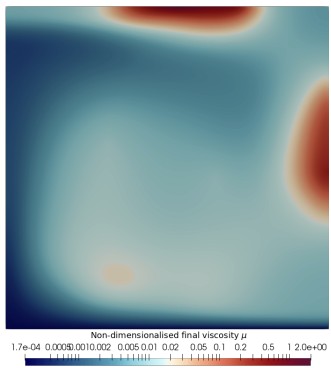
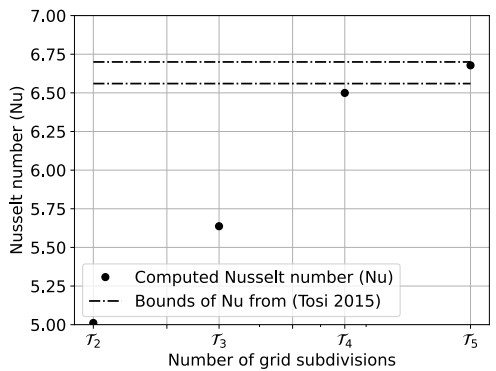

**Figure 3.** Results for the setup described in section 4.2.2: (Left) Final viscosity profile achieved after reaching a steady state with a nonlinear rheology. (Right) Convergence of the Nusselt number values from the setup from section 4.2.2 with mesh refinement.

$$\mu_{\text{linear}} = \exp(-\gamma_T T + \gamma_z z), \tag{29}$$

$$\mu_{\text{nonlinear}} = \mu^* \frac{\sigma_y}{\dot{\epsilon} : \dot{\epsilon}}, \tag{30}$$

where $\gamma_T$ and $\gamma_z$ are positive constants, and $\mu^*$ is a constant specified which is the effective viscosity in the nonlinear regime which is active when the stresses exceed the yield stress $\sigma_y$. The viscosities are considered as a $P_0$ FE function, constant per cell, and averaged to coarser grids. The FGMRES solver with a multigrid preconditioner applied to the viscous $A$-Block is used with 10 FGMRES iterations per Stokes solve and run until a single convection cell has stabilised. The nonlinear Stokes system was set to be solved with a Picard type iteration for 5 times at every time-step, although, it was sufficient to reduce after the system starts to stabilise. The final viscosity contour in $\log$ scale can be seen from fig. 3 (left). If the viscosity was only temperature-dependent, then the contours would look similar to the temperature contours. As we consider a pseudoplastic rheology, the viscosity is considerably smaller where the stresses are larger, i.e. the region where the material sinks in. This effect would also be seen where the material rises up (lower-left on the square), but there the effect aligns with the temperature dependence effect (hotter - low viscosity), whereas where material sinks down (top-right), it counters the temperature dependence. In addition, we also calculate the Nusselt number at the top boundary and note its convergence in fig. 3 (right) as we refine the domain, which progresses to the bounds reported in other codes from Tosi et al. (2015) and Davies et al. (2022).

### 4.2.3 Spherical Shell

This section involves experiments on a thick spherical shell where an incompressible Stokes system with variable viscosity coupled with the energy equation is considered (Ratcliff et al., 1996; Zhong et al., 2008; Davies et al., 2022; Euen et al., 2023). The boundary conditions prescribed are the free-slip conditions on both inner and outer surfaces. The initial condition for the temperature field is a radially linear function with an added symmetrical perturbation according to a spherical harmonic function $\mathcal{Y}_{lm}$ with degree $l$ and order $m$. The test is to check if the spherical harmonic components of the resulting temperatures



| Case | Rayleigh number (Ra) | $r_\mu$ | Perturbation symmetry |
|------|----------------------|---------|-----------------------|
| A3 | $7 \times 10^3$ | 20 | Tetrahedral |
| A7 | $7 \times 10^3$ | $10^5$ | Tetrahedral |
| C1 | $1 \times 10^5$ | 1 | Cubic |
| C3 | $1 \times 10^5$ | 30 | Cubic |

**Table 1.** Test cases considered to compare the radially averaged temperatures with published data from other codes (Euen et al., 2023)

largely maintain the prescribed order and degree. We consider two types of symmetry in perturbation of the initial temperature field: tetrahedral and cubic symmetry. The initial condition for the temperature field is given as

$$T(t = 0) = T_{\text{ref}} + \epsilon T_{\text{perturb}}, \tag{31}$$

where $\epsilon > 0$ and $T_{\text{ref}}$ is a reference profile which satisfies the steady diffusion equation. An initial perturbation of $T_{\text{perturb}} = \mathcal{Y}_{lm}$ with order $l = 3$ and degree $m = 2$ results in a tetrahedral symmetry of the plumes and downwellings, while a cubic symmetry can be constructed with $T_{\text{perturb}} = \mathcal{Y}_{40} + \frac{5}{7}\mathcal{Y}_{44}$. The prescribed viscosity variation is dependent on the evolving temperatures and is given with the following expression,

$$\mu = 10^{r_\mu\left(\frac{1}{2} - T\right)}, \tag{32}$$

where $r_\mu > 0$ and $T$ is the non-dimensional temperature whose values lie between $0$ and $1$. By increasing the value of $r_\mu$, the viscosity contrast between the plumes and downwellings can be increased. By taking a pure diffusion solution as the reference, $T_{\text{ref}} = \frac{r_{\text{min}} - r}{r_{\text{max}} - r_{\text{min}}}$, the Nusselt number can be computed with eq. (27). The experiments are performed on the thick spherical shell with $r_{\text{min}} = 1.22$ and $r_{\text{max}} = 2.22$, thereby giving the aspect ratio of the domain similar to Earth mantle $\frac{r_{\text{min}}}{r_{\text{max}}} \simeq 0.55$ while keeping non-dimensionalised mantle thickness to 1 (Ratcliff et al., 1996; Davies et al., 2022).

The Stokes system is solved with the geometric multigrid preconditioned FGMRES solver described in section 3.3. The simulations are started with 25 FGMRES steps for the first 25 time-steps, then the number of FGMRES steps are reduced to between 5 to 15 depending on the magnitude of viscosity contrasts (15 for $r_\mu > 10^2$). At each level 3 steps of the Chebyshev smoother of order 3 are applied, and MINRES is used as the coarse grid solver. The coarsest level ($\mathcal{T}_2$) of the multigrid solver contains 8 radial layers and the finest level ($\mathcal{T}_4$) contains 32 radial layers. For comparison of results with other works, we consider the scenarios C1, C3, A3 and A7 from Zhong et al. (2008) also presented in Euen et al. (2023), for which the respective values used can be seen from table 1. In addition, we perform multiple simulations and plot the variation of the Nusselt number with respect to the Rayleigh number as done in Ratcliff et al. (1996). All the simulations are performed until the system reaches a statistical steady state i.e. the radially averaged temperatures and the Nusselt numbers converge to a certain value. A CFL condition threshold (see eq. (13)) of $C_{\text{CFL}} = 1.2$ is used which the MMOC method allows, and they are gradually increased to $C_{\text{CFL}} = 1.5$ when the simulations are restarted with checkpoints. The resulting temperature isosurfaces at $T = 0.5$ coloured with velocity magnitudes for the A3 case can be seen from fig. 4 (right). The radially averaged temperatures from the final



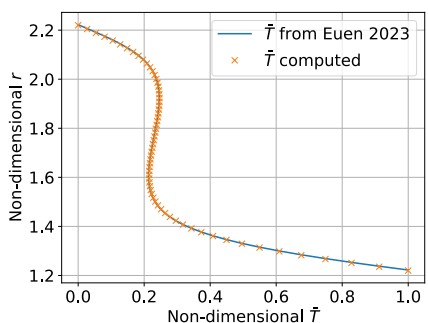
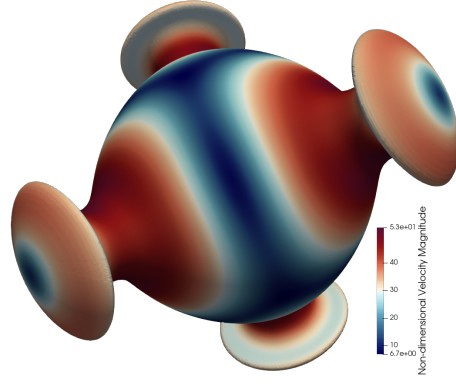

**Figure 4.** (Left) Comparison of the radially averaged Temperature values denoted by $\bar{T}$ for the A3 case from Zhong et al. (2008), plotted here with the results published from Euen et al. (2023) overlayed with results computed from our setup; (Right) Temperature isosurfaces at $T = 0.5$ for the A3 case coloured with non-dimensional velocity magnitudes.

time-step are plotted in overlay with the data presented in Euen et al. (2023) and can be seen from fig. 4 (left), and that of C1 and C3 can be seen from fig. 5 (left). The radius values from Euen et al. (2023) are scaled $(\times \frac{1}{0.45})$ for comparison with the results computed here as the mantle thickness is different in both setups, although they are comparable as the aspect ratio of the domain is the same in both setups. Nevertheless, as seen from figure, the results obtained from our framework agree well with ASPECT for these scenarios and hence, given the comparison in Euen et al. (2023), also with CitComS. For the scenario

A7, the Rayleigh number stays the same Ra$= 7 \times 10^3$ but $r_\mu$ in increased to $10^5$. This is past the transition point from a steady state regime to a stagnant lid regime. The viscosity variations are so high that the viscosity increases drastically towards the outer surface and hence no downwellings develop. The plumes that develop are localised to the lower region near the inner surface. The temperature isosurfaces are shown at $T = 0.5$ and $T = 0.8$ in fig. 6.

Next we consider multiple cases with different Rayleigh numbers and $r_\mu$ values to analyse the variation of the achieved Nusselt

number with respect to Rayleigh numbers. For a given $r_\mu$ value and a given degree $\ell$ of spherical harmonic perturbation $T_{\text{perturb}} = \mathcal{Y}_{\ell m}$, there is a specific value of Ra$_{\text{crit}}$, which indicates the onset of convection, see Zebib (1993). In this respect, we consider the Rayleigh numbers Ra$\in \{7 \times 10^3, 2 \times 10^4, 4 \times 10^4, 6 \times 10^4\}$ and that of $r_\mu \in \{3, 10, 20, 40, 100, 500, 700, 1000\}$ with cubic symmetry perturbations for all simulations. The Rayleigh numbers and $r_\mu$ values chosen are motivated from Ratcliff et al. (1996). Given that we choose the cubic symmetry for experiments, the order of the pertubation is $\ell = 4$. Hence we take

the Ra$_{\text{crit}}$ value from which is determined with the $r_\mu$ value and $\ell$ of the simulation which is available from the stability analysis of Zebib (1993) for upto $r_\mu = 1000$ and plot the achieved Nusselt numbers versus $\frac{\text{Ra}}{\text{Ra}_{\text{crit}}}$. This can be seen from figure fig. 6 (right). Each data point denotes a single simulation ran with CFL threshold of 1.2 and the Nusselt numbers are calculated at the end of the simulation. The simulations are discarded if a statistical steady state is not reached. A power law exponent of 0.31 fits the data well, which is in accordance with earlier studies (Jarvis, 1993).

Although these experiments are a challenging test to the framework, we see that the benchmarks only compare the steady state





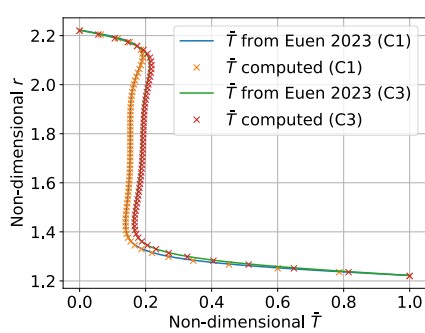
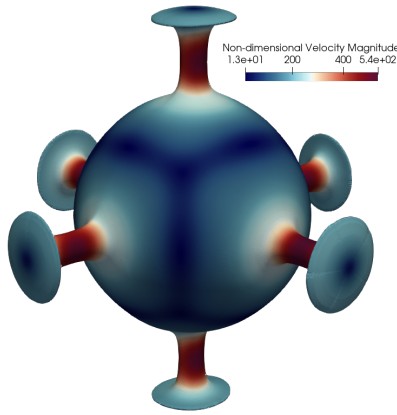

**Figure 5.** (Left) Comparison of the radially averaged Temperature values denoted by $\bar{T}$ for the C1 and C3 case from Zhong et al. (2008), plotted here with the results published from Euen et al. (2023) overlayed with results computed from our setup; (Right) Temperature isosurfaces at $T = 0.5$ for the C3 case coloured with non-dimensional velocity magnitudes

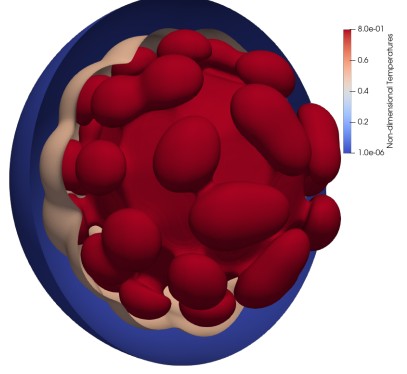
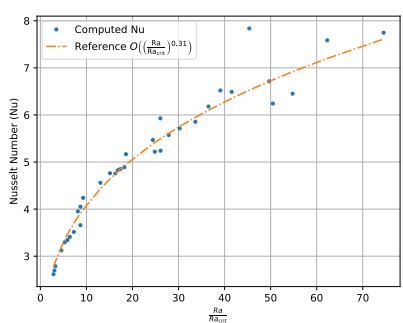

**Figure 6.** (Left) Temperature isosurfaces at $T = 0.8$ (red), $T = 0.5$ (light red) and $T = 10^{-6}$ (blue) for the A7 case from Zhong et al. (2008); (Right) Variation of Nu vs $\frac{\text{Ra}}{\text{Ra}_{\text{crit}}}$ for the cubic symmetry case, each datapoint denotes a simulation run where $\text{Ra}_{\text{crit}}$ is computed depending on the $r_\mu$ value for a given spherical harmonic perturbation, with the best fit exponent plotted side by side $\left(\frac{\text{Ra}}{\text{Ra}_{\text{crit}}}\right)^{0.31}$





result reached at the end of the simulation, while there is nothing that can be said about how the Stokes and Energy solver gets there. As the result is only verified at the end of the time evolution, the temperature and velocity patterns through which the solvers reach the solution can be different.

## 5  Scalability of the Framework

As the HYTEG framework is developed focussing towards HPC applications, scalability studies have been previously shown on test problems in, e.g., Kohl and Rüde (2022) and Böhm et al. (2025). Hence, here we consider the complexities required to succesfully run a mantle convection model on a spherical shell geometry with varying viscosity and mixed boundary conditions. This is important to highlight the abilities of the state-of-the-art framework in performing high resolution global scale modelling, and the potential optimizations that ought to be done. As the major chunk of the computational effort goes into

solving the Stokes system, that is what we will consider for the scaling experiments. The experiments are performed on the thick spherical shell $\Omega$, on which a varying viscous Stokes system is considered, eq. (5) and eq. (6).

The solver considered is the block preconditioned FGMRES with geometric multigrid on the $A$-block as described in section 3.3. The experiments were performed on the Hawk supercomputer at HLRS (66th on top500 as of Nov. 2024). We demonstrate scalability with up to approximately $10^{11}$ DoFs on $15,360$ cores. The timings reported are for 5 FGMRES steps with

restart at every 3 steps and 1 smoothing iteration in the multigrid solver. For a single weak scaling test, we fix the number of coarse mesh elements to 1 per MPI process, while increasing the number of processes and problem size respectively, maintaining the same number of DoFs per process. Each coarse mesh element is refined several times as outlined in section 2.1 and multiple weak scaling tests are performed at different $\mathcal{T}_k$. In this setup for a single weak scaling test, the highest refinement level stays the same. First a comparison between the application compiled with and without vectorization is shown in fig. 7

(left) and we achieve nearly 4x improvement with the AVX supported code generated by the HOG (Böhm et al., 2025). In the runtimes reported in fig. 7 (right), we consider multiple weak scaling cases. Each line indicates a single weak scaling test and the $\mathcal{T}_k$ next to the line denotes the highest level $(k)$ of refinement the corresponding test was performed at. From the plots, we see that the framework scales extremely well for upto 61,440 MPI processes. Lastly we demonstrate the ability of the framework to handle about $10^{11}$ DoFs in the variable viscous Stokes system on 15,360 cores. Therefore we employ a different

solver setup to reduce the memory requirements and push the size to the linear system to $10^{11}$ unknowns. The test indexed with *-R2* was performed at level $\mathcal{T}_7$ with 1 pre and post smoothing iteration per level and had a restart for FGMRES at every 2 steps slightly reducing the memory requirements thus being able to handle a system with $10^{11}$ DoFs.

To demonstrate the strong scaling of the framework, we highlight in fig. 7 (right) runs with the same problem size at different refinement levels, specifically we chose runs with $2 \times 10^9$ DoFs. The same system when solved at $\mathcal{T}_7$ on 2 nodes takes about

$\simeq 500$ seconds, while it can be brought down to $\simeq 20$ seconds on 128 nodes at refinement level $\mathcal{T}_4$. Hence these results clearly show that the framework is scalable and suitable for solving globally resolved mantle convection models.



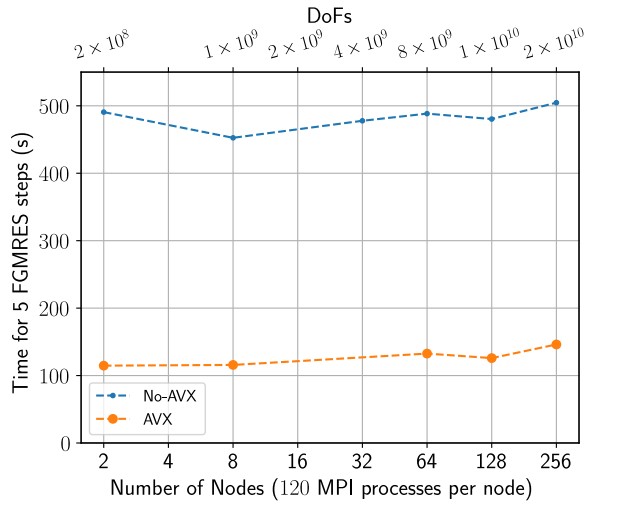
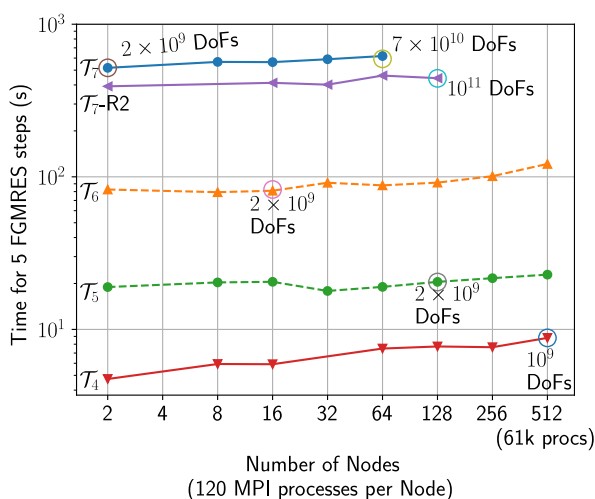

**Figure 7.** (Left) Weak scaling at $\mathcal{T}_6$ for an identical setup with and without vectorization using the code generated from the HOG generator, (Right) Weak scaling comparison with the highest multigrid level $\mathcal{T}_4$, $\mathcal{T}_5$, $\mathcal{T}_6$ and $\mathcal{T}_7$ and the index -R2 denotes that the FGMRES solver was set to restart every 2 steps (due to memory issues) in handling the variable viscous Stokes system with a peak unknowns of $10^{11}$ while showing the scalability of the framework for upto 512 nodes on Hawk supercomputer at HLRS (66th on top500 as of Nov. 2024)

## 6 Conclusions

In this paper, we presented a comprehensive evaluation of the HYTEG Finite Element framework applied to problems from computational geodynamics. We assessed the key components required for large-scale mantle convection simulations, includ-
ing the multigrid-based linear solver for the Stokes system and a semi-Lagrangian advection scheme, through a series of targeted tests and geophysical benchmarks.

We verified the expected convergence rates of the Finite Element solutions against analytical results for Stokes flow with Dirac-delta-type forcing, demonstrating the robustness of the implemented multigrid solver. We also proposed and validated a novel methodology to eliminate rigid body modes in the thick spherical shell geometry with tangential stress-free, no-normal-flow
boundary conditions.

We conducted benchmark simulations representative of realistic geodynamic scenarios, including compressible flows and non-linear, temperature-dependent rheologies. These experiments spanned from unit square domains to thick spherical shells and confirmed the physical fidelity and numerical stability of the proposed methods.

Scalability studies on the Hawk supercomputer (ranked 66th on the Top500 list as of November 2024) showed that our im-
plementation is capable of handling up to $10^{11}$ degrees of freedom in the Stokes system. While the FGMRES solver exhibits robustness under large viscosity contrasts, the convergence is hindered by inadequate coarse-grid approximations in the geometric multigrid preconditioner. Addressing this bottleneck is an ongoing area of research, with continued exploration of techniques for improved coarse-level correction.



Overall, our results demonstrate that HYTEG is a highly scalable and capable framework for highly resolved mantle convection

simulations. Future work will focus on further optimizing multigrid performance and exploring advanced solver strategies to fully exploit the potential of matrix-free methods at extreme scales.

*Code and data availability.* The software used in the experiments is available on Zenodo: 10.5281/zenodo.15276450 (Ilangovan, 2025), while the resulting data will be made accessible through the Leibniz Supercomputing Center (LRZ): 10.25927/zvnf9-cvc38 (Ilangovan et al., 2025)

*Author contributions.* PI: implementation, execution and analysis of the benchmarks, writing and editing of draft; NK: supervision, reviewing and editing of draft, acquisition of compute time; MM: supervision, reviewing and editing of draft, acquisition of compute time and funding.

*Competing interests.* The authors declare that they do not have competing interests.

*Acknowledgements.* This work was supported by the German Federal Ministry of Research, Technology and Space (BMFTR) as part of

its initiative "SCALEXA - New Methods and Technologies for Exascale Computing" (BMBF project 16ME0649 - CoMPS). Computing resources were provided by the Institute of Geophysics of LMU Munich, Oeser et al. (2006), funded by the Deutsche Forschungsgemeinschaft (DFG, German Research Foundation) – 495931446 and 518204048 and the Höchstleistungrechenzentrum Stuttgart (HLRS). The authors also wish to thank all their previous and current collaborators from the HYTEG team.



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
