# Peer review of "Highly Scalable Geodynamic Simulations with HYTEG"

_EGUsphere, 2025_

## Author Comment (AC1)

**Reply to Comments by G. Stadler**

The paper contains benchmark results obtained for the HYTEG code framework, which solves the typical equations governing mantle flows. Such comparisons and benchmarks are valuable, even if they do not add real new methods/algorithms, or new geophysics results. Generally the paper is well written, but I've a list of comments and suggestions below.

**Main comments:**

1. Line 108: I would not claim that the surface velocity components are typically taken from plate-tectonic observations. This is one option to force the system, but it should arguably be the other way round, i.e., the forces in the mantle together with the physics should result in plate-like surface velocities. Imposing the plate motion as Dirichlet bdry conditions can result in unphysical forces in the system; I suggest that you at least not claim that this is the usual/only thing done in mantle flow. It is again pointed out as the normal case in line 308/309–I dont think this is the only choice.

> **Authors' reply:** You are of course correct here in both respects. Using plate velocities is not the only option and doing that incorrectly can introduce unphysical forcing. We usually use (appropriately scaled) paleo-plate velocities in our simulations for retrodicting mantle flow in the past for the stabilising effect they have in the inversion. We will stress in the paper that this is, but only one of the possible options.

2. Eq 30, line 396 – something does not seem to be correct in the nonlinear viscosity here, there might be a missing square root. I would think that the highest power the denominator should be 1 in terms of the strain rate, corresponding to plastic yielding.

> **Authors' reply:** You are correct, apologies, the text will be corrected.

3. Regarding nonlinear Stokes solves: I assume all nonlinearity is treated with Picard fixed points iterations, which has limitations for stronger nonlinearities. Can you give some information about how many orders of viscosity variations occur in these benchmarks? Since Picard just solves a sequence of variable viscosity Stokes problems, I dont understand the comment made on line 274, namely that harmonic averages work better for nonlinear problems. These averages only affect the multigrid coarse grids, which is a purely linear issue–or am I missing something here?

> **Authors' reply:** From Fig. 3, we can see that the final non-dimensionalised viscosity changes by nearly 4 orders of magnitude across the domain. With respect to the harmonic averaging, you are fully correct. It only affects the multigrid solver. Our observation was that in this setting the multigrid solver worked better with the harmonic averaging. We will express these two points in text more clearly.

4. Rigid body modes, Sec 3.4: Makes sense to penalize rigid body modes as you discuss and thus save an MPI reduction since you avoid projecting out the rotation. It is surprising that, as discussed in Sec 4.1.2 there is sensitivity to the penalty parameter–if systems are solved reasonably accurate, one would expect that there is no sensitivity as these modes are in the null space of the PDE operator, so they should be pretty easy to control (and the solution should not degenerate if the penalty parameter is large). Does the sensitivity (and thus the need to tweak this parameter) have to do with the iterative solver?

> **Authors' reply:** Thanks for the affirmation. No, it does not have to do with the iterative solver, indeed we get the same behaviour with the direct solver as well. As this was a first test with this penalty approach, we had only presented the results that we observed.

5. Regarding scalability in Sec 5: Since you focus on the Stokes solves, the task should be the scalability of those solves, and not (only) at the scalability of a fixed number of iterations towards that solve. What you study is sometimes called the parallel scalability and it's a purely implementation issue; but it could be that more and more iterations are needed for finer discretization (in weak or even strong scaling), degrading the overall scalability of the complete solve (that second component is sometimes called algebraic scalability or simply mesh independence/robustness). For problems with severely varying coefficients (or even nonlinearities!) it's challenging and sometimes impossible to achieve that, and it requires an effective and scalable smoother, often with more than a few smoothing steps etc. This issue that two components contribute to the scalability does not come up with explicit time stepping which typically just amounts to matrix-vector applications, but it matters for implicit systems (like Stokes) and makes them much harder to scale. Please at least add a discussion of that issue to emphasize that just being able to do Krylov iterations in a scalable fashion does not necessarily imply a scalabe solver! For the larger scalability test you even reduce the number of smoothing steps and the GMRES history to save memory; I'm not convinced that you get that second part needed for overall scalability of the solver for reasonably challenging problems.

**Authors' reply:** The idea of this scalability test is indeed purely to test the implementation of the Stokes solver and how well it can cope with the increasing resolution of the models. Although we tried to mention that the solver can only 'handle' the system, it is definitely not very clear. We will add an extra discussion on nailing this point down even more in the final version. A parallel study from Burkhart et al. (2025) was also conducted which analyses the Stokes solver considering the residual reduction with large viscosity variations in the HyTeG framework.

6. It's great that you can scale to 1e11 unknowns, but it would be interesting to see settings where such a resolution is actually needed. I would guess that it only matters for extremely nonlinear rheologies, extremely varying coefficients, very fine geometric structures etc. Can you comment on your solver being used (or plans to being used) for such problems? My concern is that you vastly overresolve problems and thus do not gain any significant accurarcy by going to so many DOFs.

**Authors' reply:** As you have pointed out, for extremely nonlinear rheologies/varying coefficients, higher resolution is necessary. Although, from our tests, we see that to achieve mesh convergence even at Rayleigh numbers $\sim 10^6$, we require $\sim 25$ km resolution to capture the small scale features. A baseline case in our groups' interest is a model with a radially varying viscosity with the inclusion of an asthenosphere which introduces $\sim 10^4$ orders of viscosity changes within $\sim 150$ km. To capture the small scale dynamics and features at Earth-like Rayleigh numbers $\sim 10^7$, an ideal resolution would be with $10^{11}$ DoFs which corresponds to $\sim 6.25$ km resolution of the FE element ($\sim 3$ km resolution between grid points). A trade-off would be not to do global refinement but rather an adaptive mesh refinement. This is also an ongoing study in our framework in doing this efficiently under the matrix-free setting (Mann, B., & Rüde, U. (2025)).

**Minor:**

1. line 83: "A fact to which ..." This isn't a sentence.

2. Equation 9: The dot denotes the inner product between two vectors,
   but you also seem to use a dot for the matrix-vector multiplication
   between tau and $\hat{r}$ – is that intentional?

3. line 167: ...a brief *overview* of some...

4. line 185: a Stokes -> Stokes

5. line 308: setup more closer -> setup closer

6. line 344: in same as -> in the same as

7. line 483: upto -> up to

8. Missing commas make reading some sentences tricky–please add commas
   in the following spots (and probably some more spots):

   (a) line 100 after mantle
   (b) line 110 after no-outflow
   (c) line 148 after Here
   (d) line 197 after factor
   (e) line 216 after optimisations
   (f) line 226 after enough
   (g) line 310 after Next
   (h) line 479 after First

**Authors' reply:** Thanks for pointing these out, we will correct them in the revised version.

---

## Author Comment (AC2)

**Reply to Review by Shijie Zhong**

This paper introduced a finite element modeling framework HyTeG for modeling mantle convection. The framework appears to be very versatile with a lot of capabilities. For example, it is capable of 2-D and 3-D modeling of thermal convection with variable viscosity. It uses triangle/tetrahedral elements with quadratic/linear shape functions for velocity/pressure for numerical stability. It employs a matrix-free solver with multi-grid solver capability that does not require storage of the stiffness matrix, enabling modeling problems with extremely large number of unknowns ($10^{11}$). It uses an Eulerian-Lagrangian approach (or semi-Lagrangian method?) to solve the energy equation. The paper covers a lot of topics, as this type of papers often do, including governing equations of compressible mantle convection, numerical methods, and some benchmark results. In general, I support technical effort like what this paper presents. I can see that this paper will be eventually published, but there are a few issues I think that the authors should address and improve before it can be published.

**First, some main comments:**

1. On the benchmark. For the stationary benchmark in section 4.1 (i.e., for the Stokes flow problem), I think that the authors should present the dynamic topography and geoid benchmark results for two important reasons: a) they are geophysically important and relevant, and b) the geoid anomalies are very sensitive to the solution quality for the pressure and velocity. Analytical solutions for the geoid and dynamic topography are widely available. For example, CitcomS benchmarks in Zhong et al., [2008] have a big section on this type of benchmarks (or even in Zhong et al., [2000]). Fig. 1 for the norm-2 for flow velocity and pressure errors is encouraging, but dynamic topography and the geoid benchmarks will be much more relevant, making the code more useful and appealing to potential users.

> **Authors' reply:** Your suggestion to additionally add dynamic topography and geoid results for the verification of the code is a valid point. In the revised version, we are planning to present the response kernels for dynamic topography at the surface and CMB with respect to density perturbations (Dirac delta form of spherical harmonics) at various depths inside the mantle. As the dynamic topography and geoid are derived from the computed velocity (stress) field, we studied convergence of the radial stress on the boundary. Taking into account other comments, we are planning to add a plot presenting the order of convergence of the error in FE computed radial stress employing the CBF (Consistent Boundary Flux) method as well as an $\mathcal{L}_2$ projection of the gradient.

On the same topic, in lines 305-306, authors referenced several recent papers (since 2016) on developing semi-analytical solutions for incompressible Stokes flow problem in spherical shell using spectral methods. However, such solutions were developed in geodynamics well before 2016. For example, Tan et al. (2011) showed such analytical solutions for compressible Stokes flow in spherical shell geometry and used them to benchmark compressible version of CitcomS. Zhong et al. (2008, 2000) did the same for incompressible Stokes' flow and used them for benchmarks of incompressible CitcomS. I think that the authors should acknowledge these studies. Actually, the way that the delta function was treated in numerical benchmark calculations presented in 4.1.1 is actually the same as how it was done in CitcomS benchmark in Zhong et al., 2008. Again, some suitable reference is needed for this (see more comments on this type of issue later).

> **Authors' reply:** As we used approaches that arrive at the semi-analytical solution without the propagator matrix method, these references were unfortunately missed out. It is indeed important to mention it, and we will add these in the paper. Thanks for pointing us towards the compressible benchmark. It would definitely be a nice test for the Stokes solver in the compressible case, which we could perform for our framework, potentially in this or in a future work.

2. On the benchmark result in Figure 2's Nussult numbers vs different resolutions for 2-D compressible mantle convection, I believe that there are some errors in how King et al. (2010)'s results were presented in this Figure. The figure showed the current study's Nu's are slightly less than 7.4 at the highest resolution, but the authors presented a range of possible solutions from King et al. (2010) between 7.3 and 7.7, thus justifying their results. However, the supplementary Table S6 from King et al. (2010) showed that Nu for this case (Di=0.5 and Ra=1e5) ranges from 7.587 to 7.63 for 5 of 6 different codes, including three well-known finite element codes Citcom, Conman and UM that would be very similar to the code in this paper. Only one code in King's benchmark study had Nu at 7.50. In any case, the authors need to clarify the origin of King's benchmark results of Nu from 7.3 to 7.7. From my view, Nu=7.4 for this low Ra number case differs quite significantly from King's benchmark results, suggesting to me a concerning level of numerical error in their solutions.

**Authors' reply:** The supplementary material from King et al. (2010), unfortunately, is no longer available from the journal's webpages. Hence, we attempted to extract the values from the plot in the paper, which of course is an inexact approach. In the meantime we contacted Scott King directly, and he was so kind as to provide us with the data. We used these to update the paper. As for the argument on the values itself, the treatment of advection is quite different in our code (semi-Lagrangian). This is similar to approach used in the CZ code considered in the King benchmark. CZ was also on the lower side of the reported Nusselt numbers. Re-examining our code we also detected a minor issue in our TALA implementation. You were, thus, correct in your assessment, although even with the issue present we obtained Nusselt numbers close to the range reported. We fixed the problem and also added Nusselt number computation with the CBF method (we will be update the code version on Zenodo). We re-ran the experiment and now obtain Nusselt numbers ($\simeq 7.50 \pm 0.01$). Also velocity RMS values are very close to those for the CZ code from the benchmark.

3. For the four spherical shell convection benchmark cases listed in Table 1 that Euen et al., 2023 used for ASPECT (originally from Zhong et al., 2008), the authors should presented the steady state results of Nu, Vrms, and other quantities, as Zhong et al., (2008) and Euen et al. (2023) did. Currently, the authors only compared the temperature profiles with Euen et al. (2023) in figures. This can be improved by presenting numerical values.

**Authors' reply:** Thanks for the suggestion. We will follow it and add these values to the updated version.

4. Section 2.3.1 and 2.3.2 discussed treatments of divergence of rho*u and u for compressible convection with no reference. However, the same treatments in finite element codes for compressible mantle convection can be found in Tan et al. 2011 for spherical shell code CitcomS or Leng and Zhong for 2-D Cartesian code. The authors may want to give some references in presenting their methods.

**Authors' reply:** Thanks for pointing this out. We will add corresponding references for the frozen velocity approach. With respect to the idea of treating the full stress tensor in the momentum equations in a similar way (frozen divergence), we are, to the best of our knowledge, not aware of this having been reported on in the literature so far.

5. The authors highlighted their method as a matrix free method which they stated was the key for solving for a problem with exceedingly large numbers of unknowns. The way I see from reading this paper is that in this method, the elemental stiffness matrix Ke is generated and used to compute Ke*u without the need to assemble elemental Ke to a global stiffness matrix K and without the need to store K of course. If so, I suppose that this will significantly add to the cost of the calculation of Ku, especially if Ke needs to be formed every iteration within a given time step, if you do not store any K or Ke. I can see for constant viscosity and identical element, Ke is probably the same for all the element and only needs to be computed once. However, for variable viscosity, each element may have different Ke which would need to be computed. Therefore, in this matrix-free method, there seems to be a trade-off between computer memory saving and calculation speed. The authors may want to discuss this issue and give an order of magnitude estimate of CPU time for calculations with $10^{11}$ unknowns.

**Authors' reply:** It is well-known that compared to standard sparse matrix implementations matrix-free methods are typically more efficient in terms of memory and computational time, especially for medium- to high-order discretizations.

The memory savings are obvious, and crucial for large scale simulations. For the Stokes systems considered here, the number of non-zeros per row is in the hundreds. Adding the indexing data for sparse formats like CSR additionally doubles the memory requirements and would add a memory overhead equivalent to storing that number of additional vectors.

In terms of computational time, the matrix-free operator application is typically faster than a sparse matrix-vector product, because the latter is usually heavily memory bandwidth bound, while the matrix-free operator application tends to be compute bound for medium- to high-order discretizations and complicated bilinear forms (like the viscous terms on curved domains). This fits the performance characteristics of modern hardware much better. Studies of that are ubiquitous in the literature, see, e.g., the series of papers by Kronbichler et al. and also our study for HyTeG in Böhm et al. (2024) in SIAM J. Sci. Comp.

We will extend the discussion of the topic in the introduction accordingly.

Please also note that our problem domain is curved. This means that, even for the constant viscosity case, the local element matrices will not be identical for all elements.

The wall clock time for the $10^{11}$ DoF case is shown in Figure 9 (right).

**Some minor comments:**

1. The authors mentioned in a few places the need for mantle convection to achieve 1 km resolution. I was curious how small the time increment would have to be for such a small element, based on the criterion for picking up the time increment. Is it really feasible to use such a small time increment in global models with such a high resolution? Can a different model be formulated if 1 km is indeed needed.

> **Authors' reply:** This is a good point. We will add a comment in the paper on this. Since we use a semi-Lagrangian scheme with an implicit time discretization for the Eulerian part, we end up with a stable scheme even for relatively large timesteps. However, it is correct that we need to complement a high resolution in space with a suitable resolution in time to ensure that the time discretization error does not dominate.

2. The authors may want to double-check the writing. There are quite some grammar issues.

> **Authors' reply:** Thanks for pointing this out. We will check the paper again to improve on spelling and grammar.

3. Line 31, I am not sure if Baumgardner (1985) was for a mantle convection code – its mathematical formulation was very different from any of the mantle convection formulations we know, especially how the continuity equation was treated, as I recalled.

> **Authors' reply:** Baumgardner (1985) is, besides his PhD thesis, the original first reference for the mantle convection code TERRA. Admittedly there have been various changes to the algorithms since that time, but we think it is still a valid citation for that code. Actually, you cited the same reference, as an example for FE methods being 'widely used in the studies of mantle dynamics' in your 2007 chapter in Treatise in Geophysics.

4. For Fig. 1 and 2, can the authors explain what the resolution is for each triangulation level?

> **Authors' reply:** Your comment is correct. The refinement level $\mathcal{T}_k$ alone does not allow to derive the resolution. We will update the paper with the concrete values.

---

## Author Comment (AC3)

**Reply to Comments by Shangxin Liu**

I glimpsed this interesting geodynamic technical paper from the GMD article alert and further read it in detail. This study presents a new finite element modeling framework called HYTEG, which is based on matrix-free geometric multigrid preconditioner (like the one used by the current version of ASPECT) to overcome the need of the large memory for the storage of the stiffness matrix in the classic geometric multigrid preconditioner. The authors show the benchmark results against analytic or previous numerical codes in both 2D and 3D geometries through instantaneous and time-dependent calculations. This new framework provides a useful finite element tool for the community. While the work presented here shows the various capabilities of this new software, I have some comments and suggestions for the authors to consider to further improve the robustness of this study and the HYTEG framework.

1. While the mesh refinement is introduced in the main text, I still found that the number of mesh of each refinement level is not very clear. What are the numbers of the radial elements in each triangulation (refinement) level in each convergence plot for the errors, such as Figs. 1, 2, and 3? For example, in ASPECT, global refinement n means that there is $2^{(n+1)}$ number of radial elements in the 3D spherical shell in the default setting. It's worthwhile to describe the mesh refinement in a clearer way for the convenience of readers of this paper and potential users of the HYTEG framework.

> **Authors' reply:** Thanks for pointing out this shortcoming. We will make sure to improve the paper in this respect.

2. The mantle response (delta) function benchmark in section 4.1.

First, the authors should make it clear that how the velocity and pressure errors presented (Fig. 1) are calculated. Are they the errors of the averaged velocity and (dynamic) pressure errors of the whole domain? Which wavelengths (spherical harmonic degree and orders) are presented? These are not quite clear when I read through this part. Although the details may be presented in the earlier studies the authors refer to, I think it's worthwhile to clarify these details in the main text of the paper as well.

> **Authors' reply:** The error presented is the $\mathcal{L}_2$ norm of the error of the corresponding field over the complete domain. In the delta function cases the surface is considered separately for this. We will clarify this and the spherical harmonic degree and orders in the text.

Second, only the velocity and pressure solutions are shown. I strongly suggest the authors also calculate the responses of the surface and CMB dynamic topography and geoid. The calculation of the dynamic topography includes the radial derivatives of the velocity, which requires second-order accuracy of the velocity solutions. The geoid solutions are even more sensitive to the computational accuracy because of the counterbalance effect between the buoyancy from density integral and the dynamic topography. If the response functions of the surface and CMB dynamic topography and geoid are also shown to be accurate, the robustness of the Stokes solver for this code can be verified completely.

> **Authors' reply:** You raise a valid point. Dynamic topography and the geoid are indeed important targets for geophysical inference. In the revised version of the paper, we are planning to show the response functions of surface and CMB topography in the presence of a density anomaly in the dirac delta form of a single spherical harmonic at various depths in the mantle. Also as both of these quantities are computed from the radial stress we are also planning to show the error convergence of the FE computed radial stress to the analytical solution.

Third, at lines 305-306, several previous efforts in the formulation of the response function benchmark in 3D spherical shell geometry are referenced. However, other previous relevant studies on the same topic by the peers should also be acknowledged as well in this paper. For example, Liu and King, 2019 systematically benchmarked the Stokes solver for the open-source community code ASPECT in 3D spherical shell domain using the similar approach, following the formulation of the earlier work of Zhong et al., 2008 for another popular community code CitcomS. It's noteworthy that Zhong et al., 2008 and Liu and King, 2019 both calculated the response functions not only in isoviscous Stokes system, but also in two-layer viscosity profile with a stiff top lid. I suggest that the authors also calculate the response functions in a two-layer viscosity profile to show that the Stokes solvers of the HYTEG is able to properly handle the radial viscosity jump.

The calculation of the dynamic topography may also involve the use of consistent-boundary-flux (CBF) method to help improve the accuracy of the stress compared with the straightforward pressure smoothing method (Zhong et al., 1993). Including the effects of self-gravity will also significantly change the long-wavelength dynamic topography and geoid. The incorporation of CBF method and self-gravity into a finite element code will require considerable extra work. If not yet, I don't intend to push the authors to add them into HYTEG for this paper, but it would be necessary to make it clear in the main text that whether CBF method and self-gravity has been included in the calculation of the dynamic topography and geoid

solutions. Zhong et al., 2008 and Liu and King, 2019 use CBF method to calculate dynamic topography. The two studies present the response function benchmark for Citcoms and ASPECT for both cases with and without self-gravity.

Liu, S., & King, S. D. (2019). A benchmark study of incompressible Stokes flow in a 3-D spherical shell using ASPECT. Geophysical Journal International, 217(1), 650-667.

> **Authors' reply:** Thanks for pointing us to the CBF method for computing gradients on the boundaries and the consideration. As the HYTEG framework is capable of performing surface integrals in a matrix-free fashion, this is actually straightforward to implement in our code. Hence we have used the same for the computation of the radial stress on the surface and CMB for computing the tpography response functions. In addition, we have made an error convergence comparison between computing the gradients with the CBF method and a simple $\mathcal{L}_2$ projection. We will also add additional and relevant references.

3. Time-dependent thermal convection benchmark in 3D spherical shell.

   It's nice to see a match of the average temperature profiles between HYTEG and our recent study presented in Euen et al., 2023. However, to form a complete evaluation of the performance in 3D spherical shell thermal convection, it's necessary to also show the comparison of the RMS velocity profiles, Nusselt numbers at the top, and Nusselt numbers at the bottom between HYTEG and Euen et al., 2023. The Nusselt numbers require the calculation of the heat flux, i.e., temperature gradient. This diagnostic is a better criterion to evaluate the second-order accuracy of the temperature field. Again, CBF method can improve the accuracy of the heat flux calculation as well (Gresho et al., 1987). Whether CBF method is used in the calculation of heat flux needs to be clarified. The heat flux calculated in the Nusselt numbers of Euen et al., 2023 use the CBF algorithm incorporated into the early version of ASPECT.

   > **Authors' reply:** In the revised version, we will present a comparison of Nusselt numbers and the velocity RMS profiles with the data from Euen et al. (2023). For the heat flux calculation, the CBF method is even easier to implement than for computing pointwise variables, for which one needs to solve a system with the mass matrix. We will clarify this in text also.

4. Lines 270-278. The authors talked about the use of the element-wise viscosity averaging for the matrix-free geometric multigrid method. It would be better to further strengthen the purpose of using the element-wide viscosity averaging for this method. It will especially reduce the memory needed for largely variable viscosity cases compared with the same cases without viscosity averaging. This is similar to the handling of this problem in the current version of ASPECT code (Clevenger and Heister, 2021). In addition, the reason that why harmonic averaging works better than arithmetic averaging in nonlinear rheology needs to be specified.

   > **Authors' reply:** You are correct in pointing out the memory savings when using a $P_0$ type averaging for the viscosity. Although in practice we see that the major memory overhead is dominated by the velocity/pressure functions (and temporaries) one requires for the Stokes solver. In our work and as you mentioned, has been pointed out in Clevenger and Heister (2021), that the viscosity averaging mainly helps to improve the multigrid solver convergence. Our assumption for why we observed a better performance of harmonic averaging for this nonlinear rheology case could be that the coarse grid approximation is more accurate for our discretization. We will make this clear in text.

Minor comments

1. Line 272, computing the an average -> computing an average

   > **Authors' reply:** We will update this.

2. Euen 2023 -> Euen et al., 2023. This issue appears in some places, such as Figs. 4 and 5.

   > **Authors' reply:** We will update the legends of the figures accordingly.

3. From the equations (5) and (6), it appears that HYTEG solves the Stokes system for dynamic pressure instead of total pressure. I suggest making this point clear in the main text. For example, are the pressure terms in the later response function benchmark in section 4.1 the dynamic pressure or the total pressure?

   > **Authors' reply:** You noted this correctly. We can use different approaches here. We will better explain which one was used for which individual benchmark.